# Absence of toxin gene transfer from *Clostridioides difficile* strain 630Δ*erm* to nontoxigenic *C. difficile* strain NTCD-M3r in filter mating experiments

**Susan P. Sambol[1,2], Stuart Johnson[1,2], Adam Cheknis[1], Dale N. Gerding[1]***

**1** Edward Hines, Jr. VA Hospital, Hines, Illinois, United States of America, **2** Loyola University Chicago Stritch School of Medicine, Maywood, Illinois, United States of America

* dale.gerding2@va.gov

**Data Availability Statement:** All relevant data are within the manuscript and its Supporting information files.

## Abstract

Nontoxigenic *Clostridioides difficile* strain M3 (NTCD-M3) protects hamsters and humans against *C. difficile* infection. Transfer in vitro of the pathogenicity locus (PaLoc) to nontoxigenic strain CD37 has been reported. We repeated these conjugations using toxigenic strain 630Δ*erm* as donor and NTCD-M3 and CD37 as recipients. In order to conduct these matings we induced rifampin resistance (50ug/ml) in NTCD-M3 by serial passage on rifampin-containing media to obtain strain NTCD-M3r. 630Δ*erm*/CD37 matings produced 21 PaLoc transconjugants in 5.5 x 10⁹ recipient CFUs; a frequency of 3.8 x 10⁻⁹. All transconjugants carried the *tcd*B gene and produced toxin. 630Δ*erm*/NTCD-M3r matings produced no transconjugants in 5 assays with a total of 9.4 x 10⁹ NTCD-M3r recipient cells. Toxin gene transfer to NTCD-M3r could not be demonstrated under conditions that demonstrated transfer to strain CD37.

## Introduction

*Clostridioides difficile* is the leading cause of healthcare-acquired infection in the United States, with an estimated 29,000 deaths/yr [1, 2]. A further complication of *C. difficile* infection (CDI) is the high rate of CDI recurrence: 20%-49% in a large clinical trial, emphasizing the importance of prevention of recurrence [3]. One likely cause of CDI recurrence is disruption of normal gut microbiota caused by antibiotics [4]. A promising method of protection against CDI, both primary and recurrent episodes, is the use of non-toxigenic *C. difficile* (NTCD) to colonize the disrupted gut after antibiotic treatment and prevent colonization by toxigenic *C. difficile*. This preventive effect of NTCD has been reported since the 1980's in hamsters [5] and humans [6].

One strain of nontoxigenic *C. difficile*, restriction endonuclease analysis (REA) type M3 (NTCD-M3), has shown a high rate of success in preventing CDI. M3 is a member of the REA M group which is PCR ribotype 10. We have not done MLST typing but others have shown PCR ribotype 10 to be ST 15 [7]. The REA M group consists only of non-toxigenic strains. In

**Funding:** Funded by the US Department of Veterans Affairs Research Service under the 2013 William S. Middleton Award to DNG. The funding agency played no role in the study design, data collection and analysis, decision to publish or preparation of the manuscript.

**Competing interests:** DNG holds technology for the use of NTCD for prevention and treatment of CDI licensed to Destiny Pharma plc, Brighton, England. SPS, SJ, and AC report no competing conflicts of interest.

the hamster model, NTCD-M3 prevented CDI in colonized animals when challenged with toxigenic C. difficile strains [8]. In a Phase 2 clinical trial, patients colonized with NTCD-M3 showed a greatly reduced incidence of CDI recurrence (2%) when compared to patients who did not colonize when given NTCD-M3 (31%) [9]. These data support the use of NTCD-M3 as a biotherapeutic for the prevention of CDI and recurrent CDI.

A prior study demonstrated that in an in vitro setting, the entire pathogenicity locus (PaLoc) of toxigenic *C. difficile* strain 630Δ*erm* was transferred to non-toxigenic strain CD37 and two additional nontoxigenic *C. difficile* strains [10]. These data led the investigators to conclude that passive transfer of toxin genes from toxigenic *C. difficile* to NTCD strains in vivo could compromise the clinical effectiveness and safety of colonization with NTCD.

To assess the possibility of in vitro passive transfer of toxin genes to NTCD-M3, we utilized the methods of Brouwer et al. [10] and replicated their passive transfer experiments using toxigenic donor strain 630Δ*erm* and non-toxigenic recipient strain CD37, and then substituted NTCD-M3r as the nontoxigenic recipient strain.

## Materials and methods

### *C. difficile* strains

Toxigenic strain 630Δ*erm tcd*A:*erm*(B), containing an *erm*B gene inserted within *tcd*A (630Δ*tcd*A), toxigenic 630Δ*erm tcd*B:*erm*(B), containing an *erm*B gene inserted within *tcd*B (630Δ*tcd*B) and non-toxigenic strain CD37 (PCR ribotype 009) were donated by Dr. Sarah Kuehne (University of Birmingham, Birmingham, UK) and Dr. Peter Mullany (University College, London, UK), respectively. Strain 630Δ*erm* was typed by REA [11] as toxigenic REA type R30. Strain CD37 was typed as REA type T18, a new member of the nontoxigenic REA group T. NTCD-M3 was isolated in our lab in 1987 [8]. Rifampicin–resistant mutants of NTCD- M3 (original rifampicin MIC 0.25μg/ml) were generated by serial streaking onto taurocholate-fructose agar (TFA) plates [12] containing increasing amounts of rifampicin (Sigma-Aldrich, St. Louis, MO) from 0.5μg/ml to 50μg/ml. NTCD-M3 resistant to rifampicin at 50μg/ml was designated as isolate 6935 (NTCD-M3r). Antibiotic resistance of *C. difficile* strains used in mating experiments are shown in Table 1.

### Filter matings

Toxigenic CD strains 630Δ*tcd*A or 630Δ*tcd*B and non-toxigenic strains CD37 and rifampicin-resistant NTCD-M3r were grown overnight in Trypticase Soy Broth (TSB), (BD Difco, Fisher Scientific, Pittsburgh, PA). Titers were measured by ten-fold serial dilution on TFA plates, both plain TFA and TFA containing erythromycin (Sigma-Aldrich, St. Louis, MO), 10μg/ml for strain 630Δerm, or rifampicin 25μg/ml for CD37 and NTCD-M3r. The overnight cultures were centrifuged, and pellets resuspended in 200μl of Brain-Heart Infusion (BHI) broth (BD Difco, Fisher Scientific, Pittsburgh, PA).

**Table 1. Antibiotic resistance characteristics of *C. difficile* strains used in passive transfer experiments.**

| Strain | Alias | Toxigenicity | REA Type | Erythromycin | Rifampicin |
|---|---|---|---|---|---|
| 630Δ*erm tcd*A:*erm*(B) | 630Δ*tcd*A | Toxigenic | R24 | resistant | susceptible |
| 630Δ*erm tcd*B:*erm*(B) | 630Δ*tcd*B | Toxigenic | R24 | resistant | susceptible |
| CD37 | CD37/T18 | Nontoxigenic | T18 | susceptible | resistant |
| Wild type M3 1413 | NTCD-M3 | Nontoxigenic | M3 | susceptible | susceptible |
| NTCD-M3 6935 | NTCD-M3r | Nontoxigenic | M3 | susceptible | resistant |

The resuspended pellets of donor toxigenic strain and recipient non-toxigenic strain were combined in a single tube, then pipetted onto a sterile 0.45μm nitrocellulose filter (Cytiva Whatman, Fisher Scientific, Pittsburgh, PA) that had been placed on BHI agar plates and were incubated 24 hours at 37˚C anaerobically. Filters were placed in sterile 150mm petri dishes and washed repeatedly with 2 ml of sterile BHI pipetted over the filter surface. The cell wash solutions were plated onto selective BHI agar plates containing 5% defibrinated horse blood (Thermo Scientific Remel, Fisher Scientific, Pittsburgh, PA), erythromycin 10μg/ml, and rifampicin 25μg/ml at 100μl per plate, and incubated anaerobically for 24 hours at 37˚C.

Colonies were counted, isolated, and analyzed for transconjugation.

## DNA isolation and REA typing

Individual CD colonies were inoculated into 20ml TSB and incubated overnight at 37˚C anaerobically. Cultures were centrifuged, and cell pellets treated with the guanidine-EDTA-Sarkosyl method of DNA isolation [10]. Purified DNA was dried in a Speed-Vac vacuum concentrator (Thermo-Fisher, Grand Island, NY), then resuspended in 25μl of sterile Tris-EDTA buffer, pH 7.8. Resuspended DNA (5μl) was placed in a separate tube for subsequent PCR studies.

The remaining 20μl of purified DNA was digested with *Hin*dIII restriction enzyme [10] and the restriction fragments separated on a 0.7% agarose gel (Lonza SeaKem GTG, Fisher Scientific, Pittsburgh, PA). Restriction patterns were visually compared to established REA types in our collection and to the known REA patterns of strain 630Δ*tcd*A and 630Δ*tcd*B (REA type R30), strain CD37 (REA type T18) and REA type M3. REA groups are defined as restriction patterns with ≥90% similarity (letter designation). REA types have indistinguishable restriction patterns (numerical designations).

## PCR for *tcd*A and *tcd*B

Purified DNA isolated in the previous step was quantitated by nanodrop spectrometer at 260 nm, purity confirmed at 1.9 to 2.0 in 260/280 ratio, then diluted to a working concentration of 50 ng/μl. DNA from donor strains 630Δ*tcd*A and 630Δ*tcd*B, and from recipient strains CD37/T18 and NTCD-M3r were used as positive and negative controls for *tcd*A and *tcd*B genes respectively. DNA from putative transconjugants was tested for the presence of *tcd*A or *tcd*B using the primers in Table 2 (Eurofins Operon, Louisville, KY). Template DNA (200ng) was amplified in a mixture of 10mM Tris-Cl (pH 8.3), 50mM KCl, 0.01% gelatin, deoxynucleoside triphosphates (200mM each), primers (10pmol each), 1.8U AmpliTaq DNA polymerase (Thermo-Fisher Scientific, Grand Island, NY), and 0.15U *Pfu* DNA polymerase (Agilent, Santa Clara, CA) in the presence of 4.0mM MgCl2. Each cycle consisted of denaturation at 94˚C, annealing at the melting temperature for each primer minus 5˚C, and extension at 68˚C for 2 min per kb of amplicon, with a total of 30 cycles per PCR assay. Amplicons were run on a

**Table 2. Primers for PCR of *tcd*A and *tcd*B sequences in transconjugants.**

| Primer Name | Gene | Upstream or downstream | Primer Sequence | Amplicon size |
|---|---|---|---|---|
| 3pB-B | *tcd*B, 3' end | upstream | GATGATAGTAAGCCTTCATTTG | 2603 bp |
| 3pB-D | *tcd*B, 3' end | downstream | CTATTCACTAATCACTAATTGAG | |
| Nested primer 3pB-nu | *tcd*B, 3' end | upstream | CACCTTCATATTATGAGGATGG | 1004 bp |
| Nested primer 3pB-nd | *tcd*B, 3' end | downstream | CAGAGTCAGAGAAGTAGAAGAC | |
| A-u2 | *tcd*A, 3' end | upstream | AATGAGTACTACCCTGAGA | 3034 bp |
| A-d1-b | *tcd*A, 3' end | downstream | AATTTCTTAGTAGCACAGGAAT | |

0.9% agarose gel (Lonza SeaKem GTG, Fisher Scientific, Pittsburgh, PA) and analyzed for size against a 1Kb Plus DNA ladder (Thermo-Fisher Scientific, Grand Island, NY).

## Confirmation of *tcd*B using nested PCR primers

Purified DNA isolated in the previous step was quantitated by nanodrop spectrometer at 260/280 nm, then diluted to a working concentration of 50 ng/µl. DNA from donor strains 630Δerm *tcd*A:*erm*B, and from recipient strains CD37 and NTCD-M3r were used as positive and negative controls for *tcd*B genes respectively. DNA from putative transconjugants was tested for the presence or absence of *tcd*B using nested primers (Table 2). In the nested primer PCR, 200 ng of template DNA were amplified under the same conditions as the original PCR with the following differences: nested primers 3pB-nu and 3pB-nd were used, the positive control was strain 630Δerm *tcd*A:*erm*B, negative control was parent strain CD37, and the amplicons were run on a 0.9% agarose gel and analyzed against a 100 bp DNA ladder (New England Biolabs, Ipswich, MA).

## Toxin assays

**Cytotoxicity assay.** Transconjugant colonies and parent donor strains 630Δ*tcd*A or 630Δ*tcd*B, and parent recipient strains CD37/T18 and NTCD-M3r were inoculated into 20ml of BHI broth and incubated anaerobically for 48–72 hours. Cells were separated by centrifugation and the supernatants added to sterile 1.5ml Eppendorf microcentrifuge tubes. Supernatants were centrifuged at 16,000 x g and added to the Bartels *Clostridium difficile* Cytotoxicity Assay Kit (Trinity Biotech, Jamestown, NY), containing human fibroblast cells. Supernatants, either neat or pre-incubated with *C. difficile* toxin B neutralizing antibody for 40 minutes, were added to the test wells, and cell morphology (rounding of spindle-shaped fibroblasts) assessed at 24h and 48h anaerobic incubation at 36˚C.

**Enzyme Immunoassay (EIA).** Supernatants from transconjugant colonies and parent donor and recipient strains were generated using the same techniques as for the cytotoxicity assay. Quantitative toxin levels were determined in the transconjugant and parent donor strain supernatants by toxin EIA using *C. difficile* Tox A/B II kit (TechLab, Blacksburg, VA), and the results analyzed on an iMark Microplate Absorbance reader (Bio-Rad, Hercules, CA) at 450nm.

## DNA sequencing

DNA was prepared in our laboratory for strains NTCD-M3 and NTCD-M3r and sent to CosmosID, Rockville, MD for sequencing. Isolated genomic DNA was quantified with Qubit 2.0 DNA HS Assay (ThermoFisher, Massachusetts, USA) and quality assessed by Tapestation genomic DNA Assay (Agilent Technologies, California, USA). Library preparation was performed using KAPA Hyper Prep kit without PCR (Roche, Indianapolis, USA) following the manufacturer's recommendations. Following end repair and ligation with KAPA Unique dual adaptors containing Illumina® 8-nt dual-indices, the final libraries were purified using SPRI beads (Beckman Coulter, Indianapolis, USA). Library quality and quantity were assessed with Qubit 2.0 DNA HS Assay as well as Tapestation High Sensitivity D1000 Assay (Agilent Technologies, California, USA). Final libraries were quantified using the QuantStudio® 5 System (Applied Biosystems, California, USA) prior to equimolar pooling based on qPCR QC values. Sequencing was performed on an Illumina® NovaSeq (Illumina, California, USA) with a read length configuration of 150 PE for 6.67M PE reads (3.335 M in each direction) per sample.

The SNP's were identified through Snippy by a comparison of short read data and the draft assemblies. The genes were then identified through genome annotation using Prokka in

conjunction with Snippy. The SNP's were validated through a series of alignments between short read data and the draft assemblies.

## Results

### PaLoc transfer

Transconjugants were defined as *C. difficile* colonies growing on selective BHI plates containing erythromycin at 10μg/ml and rifampicin at 25μg/ml that were identified by REA typing as nontoxigenic strains and shown to contain PaLoc *tcd*A or *tcd*B genes by PCR. Transconjugation was further confirmed by toxin assays performed on transconjugant supernatants. Three filter matings were conducted between strain 630Δ*tcd*A and strain CD37/T18, generating 21 transconjugants from a total of $5.5 \times 10^9$ non-toxigenic cells, a passive transfer frequency of $3.8 \times 10^{-9}$ CD37 cells (s.d. = $4.7 \times 10^{-9}$) (Table 3). Four filter matings were conducted between strain 630Δ*tcd*A and NTCD-M3r (resistant to rifampicin 50ug/ml), generating no transconjugants from a total of $7.9 \times 10^9$ non-toxigenic cells, at a passive transfer frequency of 0 transconjugants per $10^9$ NTCD-M3r cells (Table 3). One filter mating was conducted between strain 630Δ*tcd*B and non-toxigenic NTCD-M3r, generating no transconjugants from a total of $1.46 \times 10^9$ non-toxigenic cells, at a passive transfer frequency of 0 transconjugants per $10^9$ NTCD-M3r cells (Table 3).

### REA typing of transconjugants

Toxigenic strains 630Δ*tcd*A and 630Δ*tcd*B were both identical REA type R30. REA designated CD37 as type T18. NTCD-M3r was reconfirmed as REA Type M3. There were no NTCD-M3r transconjugants, so all analyses were performed only on the 21 transconjugants generated from 3 filter matings of 630Δ*tcd*A with CD37/T18. CD37/T18 transconjugants displayed multiple variations in the REA pattern of CD37/T18 but retained 90% homology with the parent T18 type as described by Clabots et al. [11], keeping the transconjugants within the REA T group (Fig 1). It is likely that the extra and missing bands of REA type T18 are due to the inclusion of *Hin*dIII sites in the passive transfer of large segments of 630Δ*tcd*A DNA within or flanking the PaLoc, as described by Brouwer et al. [10].

### PCR analysis of transconjugants

All 21 CD37 transconjugants showed positive amplification of a 2600 bp fragment of the 3' end of the *tcd*B gene in the PaLoc, confirming the transfer of PaLoc sequences to the non-toxigenic strain (Fig 2). The amplicon size in the transconjugants matched the amplicon size of the control strain 630Δ*tcd*A. Similarly, all 21 transconjugants demonstrated amplification of the

**Table 3. Filter matings between toxigenic strain 630Δ*tcd*A and 630Δ*tcd*B and nontoxigenic strains CD37/T18 and NTCD-M3r.**

| Mating no. | Donor strain titer | Total cfu of donor strain | Recipient strain Titer | Total cfu of recipient strain | Results |
|---|---|---|---|---|---|
| 1 | 630ΔtcdA .85 x 10⁸ cfu/ml | $1.7 \times 10^9$ | CD 37/T18 0.53 x 10⁸ cfu/ml | $1.0 \times 10^9$ | 12 CD37 transconjugants |
| 2 | 630ΔtcdA .85 x 10⁸ cfu/ml | $1.7 \times 10^9$ | NTCD-M3r 0.97 x 10⁸ cfu/ml | $1.94 \times 10^9$ | 0 M3r transconjugants |
| 3 | 630ΔtcdA 1.05 x 10⁸ cfu/ml | $2.1 \times 10^9$ | NTCD-M3r 0.73 x 10⁸ cfu/ml | $1.46 \times 10^9$ | 0 M3r transconjugants |
| 4 | 630ΔtcdB 1.6 x 10⁸ cfu/ml | $3.2 \times 10^9$ | NTCD-M3r 0.73 x 10⁸ cfu/ml | $1.46 \times 10^9$ | 0 M3r transconjugants |
| 5 | 630ΔtcdA 1.8 x 10⁸ cfu/ml | $4.5 \times 10^9$ | CD 37/T18 1.0 x 10⁸ cfu/ml | $2.5 \times 10^9$ | 4 CD37 transconjugants |
| 6 | 630ΔtcdA 1.8 x 10⁸ cfu/ml | $4.5 \times 10^9$ | NTCD-M3r 1.4 x 10⁸ cfu/ml | $3.5 \times 10^9$ | 0 M3r transconjugants |
| 7 | 630ΔtcdA 1.8 x 10⁸ cfu/ml | $3.6 \times 10^9$ | CD 37/T18 1.0 x 10⁸ cfu/ml | $2.0 \times 10^9$ | 5 CD37 transconjugants |
| 8 | 630ΔtcdA 1.8 x 10⁸ cfu/ml | $3.6 \times 10^9$ | NTCD-M3r 0.5 x 10⁸ cfu/ml | $1.0 \times 10^9$ | 0 M3r transconjugants |

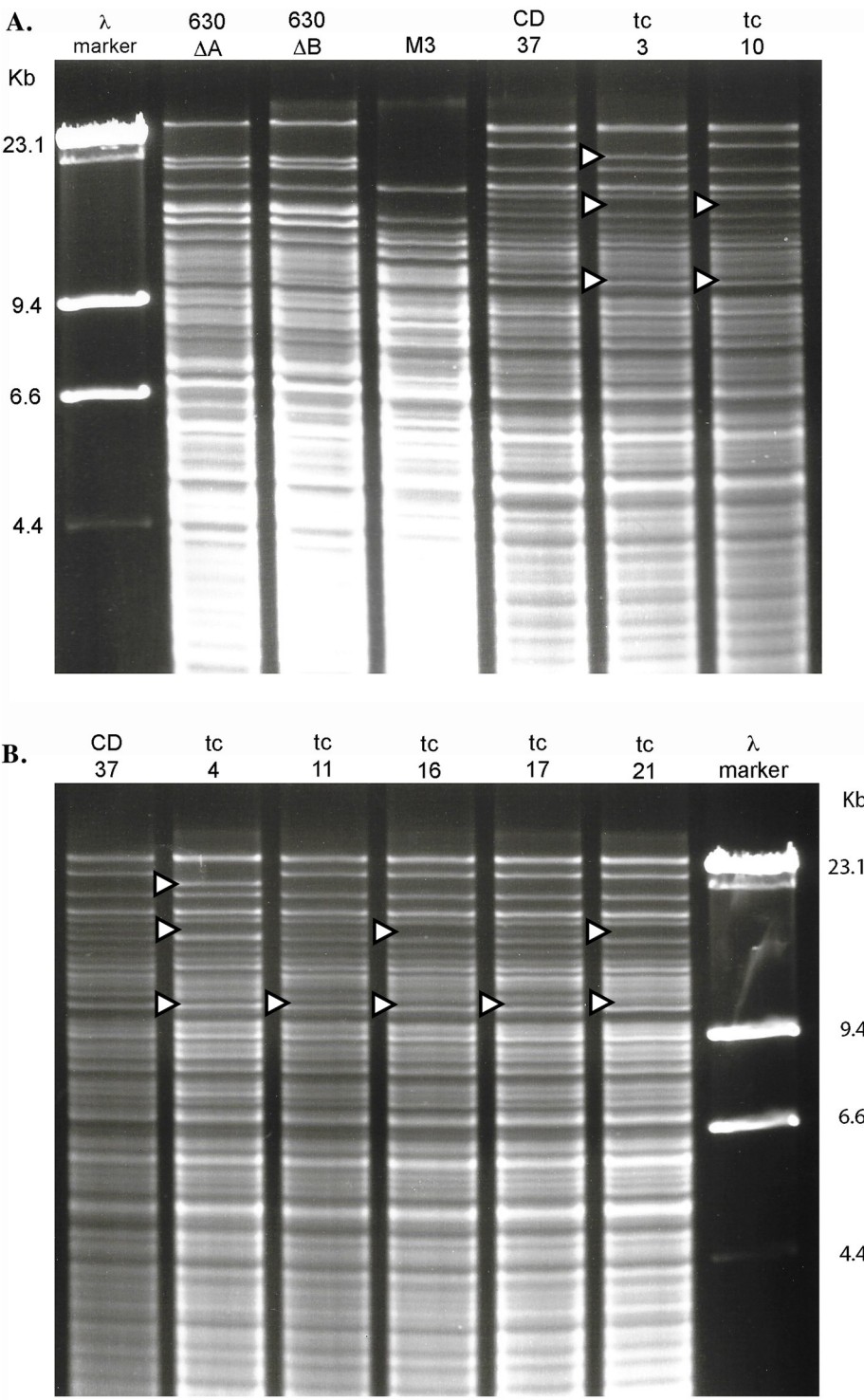

**Fig 1. *Hin*dIII REA patterns of *C. difficile* donor (Strain 630Δerm), recipient strains (M3r, CD37), and seven representative CD37 transconjugants (tc). A**. Lane 1; lambda DNA phage marker followed by donor strains 630Δ*tcd*A, 630Δ*tcd*B, M3, CD37, and transconjugants tc3 and tc10. **B**. Lane 1; CD37, followed by tc4, tc11, tc16, tc17, tc21 and lambda DNA phage marker. White arrowheads to the left of the gel lanes indicate *Hin*dIII band differences between parent strain CD37 (REA type T18) and CD37 transconjugants (tc3 –tc21).

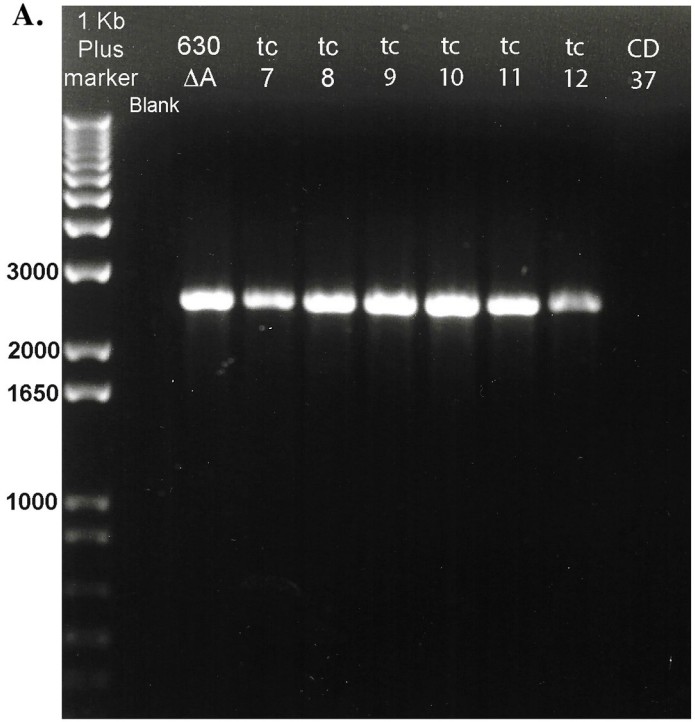

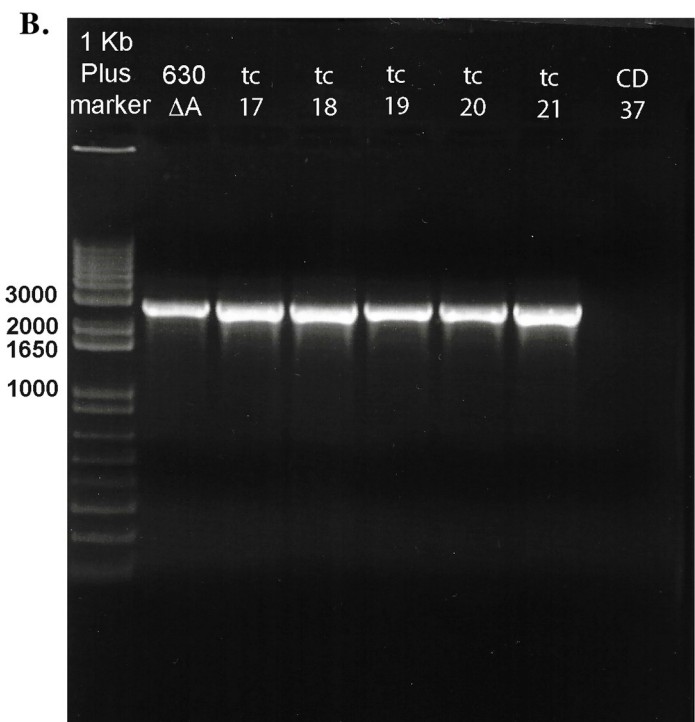

**Fig 2. PCR amplification of 3' *tcd*B sequences in 11 CD37 transconjugants and donor strain 630Δ*tcd*A. A**. Lane 1; 1Kb Plus DNA ladder with the donor strain 630ΔA in the next lane followed by transconjugants tc7- tc12 and TC37. **B**. Lane 1; 1Kb Plus DNA ladder with the donor strain 630ΔA in the next lane followed by transconjugants tc17-21 and CD37. All transconjugants show amplicons of the expected 2600bp size found in 630ΔA. The far right lanes in both gels show the negative results for recipient parent strain CD37.

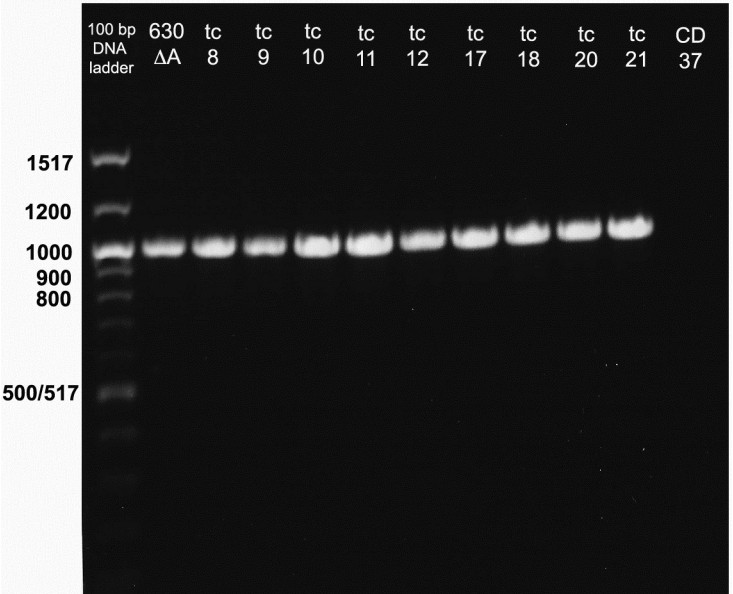

**Fig 3. PCR amplification of 3' *tcd*B nested 1004 bp sequence using primers 3pB-nu and 3pB-nd.** Lane 1; 100 bp DNA ladder with the donor strain 630ΔA in the next lane followed by 9 transconjugants (Tc 8–12, 17, 18, 20 and 21) and NTCD strain CD37 in the last lane.

1004 bp amplicon using nested primers in the *tcd*B gene. Representative transconjugants are shown in Fig 3.

## Toxin assays

Strain 630Δ*tcdA* showed cytotoxic effects (rounding of cells) identical to the Bartels kit toxin control, as did all 21 of the CD37/T18 transconjugants. Parent nontoxigenic strain CD37/T18 showed no cytotoxic effect. Specificity of cytotoxic effect was confirmed by negative results for all supernatants incubated with neutralizing antibody prior to addition to the fibroblast culture. Strain 630Δ*tcdA* and the CD37/T18 transconjugants were also positive by toxin A/B EIA (ODs >0.120 at 450nm), whereas CD37/T18 nontoxigenic parent strain gave a negative reading equivalent with background ($OD_{450}$ approximately 0.043).

## DNA sequencing of NTCD-M3 and NTCD-M3r

NTCD-M3 and NTCD-M3r Illumina DNA sequencing showed the strains differ by 2 SNPs in the core genome. One SNP is in the *pepT*_1 gene. No phenotypic changes are expected because the change results in a synonymous variant. The second SNP is in the expected *rpoB* gene which is presumably altered with rifampin resistance. Phenotypic changes are expected because this SNP results in a missense variant. Both SNPs were called with a high level of confidence.

## Discussion

Previous studies in addition to ours have shown that colonization with NTCD is effective in preventing CDI in animal models [13–15]. A surveillance study in two hospitals showed that in patients recently asymptomatically colonized with *C. difficile* (46% with a non-toxigenic strain) had a significantly lower incidence of CDI than patients who were in the same setting

and had not been previously colonized [16]. NTCD-M3 prevention of lethal infection by toxigenic strains of *C. difficile* was shown in the hamster model and prevention of recurrent CDI was shown in a Phase 2, randomized, placebo-controlled clinical trial [8, 9].

Intentional colonization with nontoxigenic *C. difficile* strains to protect against CDI could be compromised if these strains were to convert to toxin-producing strains in vivo at a substantial frequency. Brouwer et al. demonstrated spontaneous transfer of the PaLoc from toxigenic *C. difficile* strain 630Δerm to nontoxigenic *C. difficile* strains CD37, OX904, and OX2157 [9]. We confirmed that the 630Δ*tcd*A strain transferred the PaLoc to CD37 at nearly the same rate reported by Brouwer et al., generating 21 transconjugants in a total of $5.5 \times 10^9$ recipient CD37 cells, a transfer frequency of $3.8 \times 10^{-9}$. In contrast, we were unable to detect PaLoc passive transfer between 630Δ*tcd*A or 630Δ*tcd*B and NTCD-M3r using the same experimental conditions in five separate assays in over $9.4 \times 10^9$ NTCD-M3r recipient cells.

It is not clear why NTCD-M3 transconjugants were not found. There is the potential for CD37 and NTCD-M3 strain to have differences in capsule or S-layer permeability that restrict any mating apparatus [17], CRISPR systems that exclude the DNA (unlikely but possible), poor recombination in the NTCD-M3 strain compared to CD37, or CD37 is more primed to take up DNA from horizontal gene transfer. We currently favor restriction-modification system differences as the most likely mechanism since CD37 (and CD630) have both been reported to lack restriction systems [18]. However, we have no knowledge of the restriction modification systems in OX904 and OX2157 that were successfully used in transfer experiments by Brouwer et al. [10].

If a patient already harbors a toxigenic strain of *C. difficile*, is PaLoc transfer really an issue?

Should transfer occur this should not worsen the patient condition but could result in symptomatic *C. difficile* infection due to the new toxigenic NTCD strain that would require treatment. Theoretically the new toxigenic strain could possess more virulence than other toxigenic strains, but this is speculative. High frequency of such transfer events would certainly compromise use of NTCD as a preventive strategy.

Studies that attempt to demonstrate a negative, in this case, the failure of a toxigenic *C. difficile* strain to transfer its PaLoc to NTCD-M3, have unavoidable limitations. Demonstration of negative in vitro results, even in multiple assays, does not prove that PaLoc transfer from toxigenic strains to NTCD-M3 is not possible under different conditions since the mechanism of transfer resistance in NTCD-M3 has not been identified. In addition, the process of inducing rifampin resistance to obtain NTCD-M3r could have inadvertently altered DNA transfer. Moreover, the important observation of in vivo PaLoc transfer and its consequences has not been identified in animal models or humans.

## Conclusion

We conclude that the risk of PaLoc transfer to NTCD-M3 in vitro is lower than that of CD37 strain of *C. difficile* under the same conditions. PaLoc transfer in vivo has not been demonstrated nor have its consequences been determined were it to occur. The potential benefits of NTCD-M3 for prevention of primary and recurrent CDI currently outweigh the theoretical safety risk of PaLoc transfer.

## Supporting information

**S1 Raw images.**
(PDF)

## Acknowledgments

The authors thank Dr. Peter Mullany, University College, London, UK and Dr. Sarah Kuehne, School of Dentistry, Institute for Clinical Sciences and Institute for Microbiology and Infection, University of Birmingham, Birmingham, UK for their contributions of organisms used in these experiments. We also thank Dr. Pehga Johnston for technical assistance in pilot studies.

## Author Contributions

**Conceptualization:** Susan P. Sambol, Stuart Johnson, Dale N. Gerding.

**Data curation:** Susan P. Sambol, Adam Cheknis.

**Formal analysis:** Susan P. Sambol, Stuart Johnson, Dale N. Gerding.

**Funding acquisition:** Dale N. Gerding.

**Investigation:** Adam Cheknis, Dale N. Gerding.

**Methodology:** Susan P. Sambol, Stuart Johnson, Adam Cheknis, Dale N. Gerding.

**Resources:** Dale N. Gerding.

**Supervision:** Stuart Johnson, Dale N. Gerding.

**Writing – original draft:** Susan P. Sambol.

**Writing – review & editing:** Stuart Johnson, Adam Cheknis, Dale N. Gerding.

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
