## [Decision Letter · Decision Letter 0]

13 Oct 2021

PONE-D-21-28123Resistance to Toxin Gene Transfer in Nontoxigenic Clostridioides difficile Strain M3PLOS ONE

Dear Dr. Gerding,

Thank you for submitting your manuscript to PLOS ONE. After careful consideration, we feel that it has merit but does not fully meet PLOS ONE’s publication criteria as it currently stands. Therefore, we invite you to submit a revised version of the manuscript that addresses the points raised during the review process.

One expert reviewer raises concerns about the limited scope of your study and further notes that this may not be adequately reflected in the manuscript's discussion and title. I concur with this view and request that the manuscript be adjusted accordingly. In particular, the manuscript's title should be more specific about the observation made and the discussion should be expanded to include potential reasons for the lack of transconjugants.

We look forward to receiving your revised manuscript.

Kind regards,

Ulrich Nübel

Academic Editor

PLOS ONE

Journal Requirements:

[DNG holds patents and technology for the use of non-toxigenic C. difficile for prevention and treatment of CDI licensed to Destiny Pharma, plc Brighton, England. All other authors have no competing interests.] 

Reviewers' comments:

Reviewer's Responses to Questions

**Comments to the Author**

1. Is the manuscript technically sound, and do the data support the conclusions?

Reviewer #1: Yes

2. Has the statistical analysis been performed appropriately and rigorously? 

Reviewer #1: Yes

3. Have the authors made all data underlying the findings in their manuscript fully available?

Reviewer #1: No

4. Is the manuscript presented in an intelligible fashion and written in standard English?

Reviewer #1: No

5. Review Comments to the Author

Reviewer #1: In this manuscript the authors tackle an important question: can the pathogenicity locus of a toxigenic strain of C. difficile transfer to a non-toxigenic C. difficile strain that has potential as an intervention for C. difficile infection? The approaches are relevant, the data sound, and the manuscript is mostly written in a balanced way. The fact that the major finding is negative (lack of transconjugants) cannot exclude the possibility that transfers occurs under other conditions; absence of proof is no proof of absence (also stated by authors in L198-204). Overall, the manuscript is rather limited in scope and largely replicates published work in methodology (as also indicated, e.g. in L60-63); it feels more suited for a note, rather than a full manuscript. In particular, a single donor (630Derm) is used, and only one recipient (NTCD-M3r) in addition to a published control (CD37) under a single condition for transfer (filter mating). Though this does not necessarily detract from the observation that PaLoc transfer to NTCD-M3r is not found, the manuscript would be much more impactful if a) multiple toxigenic C. difficile strains were explored as donors and b) transfer under conditions more relevant to those encountered during interventions (e.g. in a hamster or mouse model of disease) could be demonstrated (at least for CD37) and authors might want to more explicitly acknowledge the narrow scope of their results.

Major comments

1. I find the title somewhat misleading and over-stating. Resistance suggests that there is an active mechanism that prevents PaLoc transfer. This cannot be concluded from the study. Moreover, the title does not reflect the limited scope of the conditions tested. I suggest rephrasing in line with the experiments performed: e.g. “No evidence for transfer of tcdA and tcdB from C. difficile strain 630Δerm to strain NTCD-M3r during filter mating experiments”.

2. The authors do not comment on whether serial streaking (in order to obtain rifR colonies) might have generated additional mutations that might affect their results. The NTCD-M3R should be sequenced and compared to NTCD-M3 to exclude secondary mutations. Similarly, in the manuscript, authors should acknowledge the possibility that results obtained with NTCD-M3r might not be different from M3 (when the Rif mutation affects transfer efficiency).

3. The results obtained with NTCD-M3r appear in contrast with those over Brouwer et al, that demonstrated PaLoc transfer to multiple NTCD strains. The current manuscript does not attempt to speculate about the reasons for this. Considering the limitations in the scope, I think this should be done. Why would NTCD-M3r behave differently from CD37, OX904, OX2157?

Minor comments

1. L48-49: Please include additional typing information on the M3 strain, in particular PCR ribotype, and MLST type. I would also like to encourage authors to further clarify whether REA-type M3 is a group that only harbors NTCD strains, or also encompasses toxigenic isolates. I also suggest citing work of others using different NTCD strains, at least in a minimal form.

2. L60: inconsistent use of italics (in vitro). Please review carefully throughout manuscript.

3. L62: authors refer to strain 630, but their experimental methods state that they used 630Δerm-derived strains. Please ensure the utmost accuracy here, as the source of 630 and 630-derived strains (such a 630Δerm and 630E/JIR8094) can result in significant genomic differences for which it is difficult to predict how these would impact the results. See for instance doi: 10.1016/j.anaerobe.2018.04.015.

4. L153-155: why were 4 matings done with one donor, and only 1 with the other? Please explain the rationale for this. The present result begs different questions: a) was this a (single) experimental failure? b) does tcdB insertion affect PaLoc transfer?

5. Erythromycin and rif resistant mutants are generally easily obtained. The authors describe a clear definition of transconjugants, but do not explicitly comment on frequencies obtained for RifR donors or EryR recipients (both not transconjugant). Were non observed? L161 and on may suggest that this is the case, but it is unclear. Please provide REA patterns for all TCs and controls, either in Figure 1 or as Supplemental information. How was the 90% homology defined? This is not clear from the M&M.

6. L188 and on; I would appreciate the author’s view on how relevant transfer of PaLoc from TCD to NTCD is, in the context of a patient. If patient already harbors a TCD, is PaLoc transfer really an issue?

7. L190: I don’t believe Brouwer et al have demonstrated that transfer is passive (nor proven that it is active).

8. Figure 2: though size of the amplicon is as expected, it would be best to confirm identity either using a nested PCR or sequencing approach.

6. PLOS authors have the option to publish the peer review history of their article (what does this mean?). If published, this will include your full peer review and any attached files.

Reviewer #1: **Yes: **Wiep Klaas Smits

---

## [Author Response · Author response to Decision Letter 0]

18 May 2022

Our responses to the Reviewer comments are shown below in red font. We believe the submission is much improved as a result of these thoughtful comments from reviewer Wiep Klaas Smits.

Reviewer #1: In this manuscript the authors tackle an important question: can the pathogenicity locus of a toxigenic strain of C. difficile transfer to a non-toxigenic C. difficile strain that has potential as an intervention for C. difficile infection? The approaches are relevant, the data sound, and the manuscript is mostly written in a balanced way. The fact that the major finding is negative (lack of transconjugants) cannot exclude the possibility that transfers occurs under other conditions; absence of proof is no proof of absence (also stated by authors in L198-204). 

Overall, the manuscript is rather limited in scope and largely replicates published work in methodology (as also indicated, e.g. in L60-63); it feels more suited for a note, rather than a full manuscript. In particular, a single donor (630Derm) is used, and only one recipient (NTCD-M3r) in addition to a published control (CD37) under a single condition for transfer (filter mating). Though this does not necessarily detract from the observation that PaLoc transfer to NTCD-M3r is not found, the manuscript would be much more impactful if a) multiple toxigenic C. difficile strains were explored as donors and b) transfer under conditions more relevant to those encountered during interventions (e.g. in a hamster or mouse model of disease) could be demonstrated (at least for CD37) and authors might want to more explicitly acknowledge the narrow scope of their results.

We agree with the narrow scope of the presentation and agree that additional donor filter matings are required even though further negative results will not resolve the question. Our purpose in this presentation was to replicate the work of Brouwer et al with known strains and a proven method of PaLoc transfer to serve as a control rather than blindly trying mating pairs and methods in vivo and in vitro. Because of the unique antibiotic resistances that must be induced or present in mating pairs, additional filter matings will require considerable time to document successful methods and obviously be limited to only another few selected isolates. We cannot agree more with the reviewer that “absence of proof is no proof of absence”.

Major comments

1. I find the title somewhat misleading and over-stating. Resistance suggests that there is an active mechanism that prevents PaLoc transfer. This cannot be concluded from the study. Moreover, the title does not reflect the limited scope of the conditions tested. I suggest rephrasing in line with the experiments performed: e.g. “No evidence for transfer of tcdA and tcdB from C. difficile strain 630Δerm to strain NTCD-M3r during filter mating experiments”.

We agree with the reviewer and have changed the title to “Absence of toxin gene transfer from Clostridioides difficile strain 630∆erm to nontoxigenic C. difficile strain NTCD-M3r in filter mating experiments”

2. The authors do not comment on whether serial streaking (in order to obtain rifR colonies) might have generated additional mutations that might affect their results. The NTCD-M3R should be sequenced and compared to NTCD-M3 to exclude secondary mutations. 

We have had NTCD-M3 and NTCD-M3r Illumina sequenced and they differ by 2 SNPs in the core genome. One SNP is in the pepT_1 gene. No phenotypic changes are expected because the change results in a synonymous variant. The second SNP was in the expected rpoB gene as we have previously reported which is presumably altered with rifampin resistance. Phenotypic changes are expected because this SNP results in a missense variant. Both SNPs were called with a high level of confidence. This is now reported in the Methods (lines 157-173) and Results (lines 239-244) of the revised submission.

Similarly, in the manuscript, authors should acknowledge the possibility that results obtained with NTCD-M3r might not be different from M3 (when the Rif mutation affects transfer efficiency).

We have now included this acknowledgment in the Discussion (lines 284-285) as a potential limitation of the study.

3. The results obtained with NTCD-M3r appear in contrast with those over Brouwer et al, that demonstrated PaLoc transfer to multiple NTCD strains. The current manuscript does not attempt to speculate about the reasons for this. Considering the limitations in the scope, I think this should be done. Why would NTCD-M3r behave differently from CD37, OX904, OX2157?

This is a challenging question particularly since the method of DNA transfer in successful strains has not been determined. We currently have no data to support any particular hypothesis but have consulted with a number of C difficile molecular experts for their views. There is the potential for restriction modification system differences between the CD37 and NTCD-M3 strain, differences in capsule or S-layer permeability (Merrigan MM et al Plos One 2013;8:e78404) that restrict any mating apparatus, CRISPR systems that exclude the DNA (unlikely but possible), poor recombination in the NTCD-M3 strain compared to CD37, or the CD37 is more primed to take DNA from horizontal gene transfer. Of these possibilities we currently favor restriction-modification system differences since CD37 (and CD630) have both been reported to lack restriction systems (Minton N et al Anaerobe 41 (2016) 104-112). However, we have no knowledge of the restriction modification systems in OX904 and OX2157. We have added this information to the Discussion as requested lines 263-271.

Minor comments

1. L48-49: Please include additional typing information on the M3 strain, in particular PCR ribotype, and MLST type. I would also like to encourage authors to further clarify whether REA-type M3 is a group that only harbors NTCD strains, or also encompasses toxigenic isolates. I also suggest citing work of others using different NTCD strains, at least in a minimal form.

The REA M group of which M3 is a member is PCR ribotype 10. We have not done MLST but others have shown PCR ribotype 10 to be ST 15. (Griffiths D et al J Clin Microbiol 2010;78:770-778.) The REA M group consists only of non-toxigenic strains. This has been added to the text lines 45-47.

2. L60: inconsistent use of italics (in vitro). Please review carefully throughout manuscript.

Thank you. We have done a search and correct on (in vivo and in vitro) and believe they are no longer italicized.

3. L62: authors refer to strain 630, but their experimental methods state that they used 630Δerm-derived strains. Please ensure the utmost accuracy here, as the source of 630 and 630-derived strains (such a 630Δerm and 630E/JIR8094) can result in significant genomic differences for which it is difficult to predict how these would impact the results. See for instance doi: 10.1016/j.anaerobe.2018.04.015.

Good point and we agree and have changed 630 to 630Δerm

4. L153-155: why were 4 matings done with one donor, and only 1 with the other? Please explain the rationale for this. The present result begs different questions: a) was this a (single) experimental failure? b) does tcdB insertion affect PaLoc transfer? 

Our major goal in these experiments was to compare PaLoc transfer to NTCD-M3 compared to CD37 and we began the experiments with the most important toxin, toxin B. After repeated attempts to show transfer with the the tcdA knockout (630∆tcdA) we made one attempt with the tcdB knockout (630∆tcdB) to get confirmation that we were not getting transconjugants. In retrospect it might have been better to do more attempts with this donor, but it was apparent that we were not getting any transconjugants.

5. Erythromycin and rif resistant mutants are generally easily obtained. The authors describe a clear definition of transconjugants, but do not explicitly comment on frequencies obtained for RifR donors or EryR recipients (both not transconjugant). Were non observed?

None were observed. 

L161 and on may suggest that this is the case, but it is unclear. Please provide REA patterns for all TCs and controls, either in Figure 1 or as Supplemental information.

We have gels of all TCs but show 7 of 21 TCs in Figure 1 as representative of all TCs, the donors and recipients. These seven TCs demonstrate the seven unique REA patterns seen in the 21 CD37 TCs; many of the TCs are identical to each other or to the parent CD37 (in which PCR and phenotypic assays show the presence of toxin B gene). Brouwer et al have demonstrated in their CD37 transfer assays that the transferred DNA fragments contain much more DNA than just the PaLoc sequences, fragments ranging from 66 Kb to 271 Kb in size. It is possible, even likely, that these additional sequences carry HindIII sites that produce additional pattern bands in the size range of 9 to 24 Kb, as seen in Figure 1. We cannot without extensive sequencing confirm that these changes in the REA pattern are due to the transfer of additional DNA with HindIII restriction sites that result in the changes and do not agree that showing all 21 transconjugants would make this any more convincing. 

How was the 90% homology defined? This is not clear from the M&M.

REA profile interpretation is from Clabots et al Ref 10. REA types were organized into larger groups on the basis of band pattern similarity. Similarities between the new and reference REA types were scored by visual comparison of each 1-mm segment of the top 60 mm of the DNA band patterns run on the same gel (approximate molecular weight range, 30 to 2 kb). A similarity index was calculated as the number of identical segments expressed as a percentage of the total segments. Any new REA type with a similarity index of >90% compared with an existing reference REA type was included in that group. New REA types were run on the same gel with a reference REA type from one to eight groups that most closely matched the DNA band pattern of the new isolate. Any new REA type with a similarity index of <90% compared with any existing reference REA types was designated the primary reference REA type for a new group and was used for all future group comparisons. The groups were designated by letters, and distinct REA types within a group were designated by Arabic numbers. We have now included ref 10 in the Results Section line 198 so the reader can reference the method.

6. L188 and on; I would appreciate the author’s view on how relevant transfer of PaLoc from TCD to NTCD is, in the context of a patient. If patient already harbors a TCD, is PaLoc transfer really an issue?

The reviewer is correct that this should not worsen the patient condition but could result in symptomatic C difficile infection due to the NTCD strain that would require treatment. Theoretically the new toxigenic strain could possess more virulence than other toxigenic strains, but this is all quite speculative. High frequency of such transfer events would certainly compromise use of NTCD as a preventive strategy. We have added a paragraph to the Discussion lines 273-278.

7. L190: I don’t believe Brouwer et al have demonstrated that transfer is passive (nor proven that it is active).

We agree with the Reviewer and have deleted the word passive.

8. Figure 2: though size of the amplicon is as expected, it would be best to confirm identity either using a nested PCR or sequencing approach.

We have now used a nested PCR to confirm the identity of the tcdB sequence as suggested. This procedure was added to the Methods (lines 129-138) and Results (lines 214-216) and is shown in new Figure 3. Nested primers are now added to Table 2.

We have also added a sentence to the Abstract to explain how NTCD-M3r was obtained from NTCD-M3, lines 22-24.

---

## [Editor Report · Decision Letter 1]

6 Jun 2022

Absence of toxin gene transfer from Clostridioides difficile strain 630∆erm to nontoxigenic C. difficile strain NTCD-M3r in filter mating experiments

PONE-D-21-28123R1

Dear Dr. Gerding,

Thank you for the careful and thorough revision of your manuscript. We’re pleased to inform you that your manuscript has been judged scientifically suitable for publication and will be formally accepted for publication once it meets all outstanding technical requirements.

Kind regards,

Ulrich Nübel

Academic Editor

PLOS ONE
---

## [Editor Report · Acceptance letter]

17 Jun 2022

PONE-D-21-28123R1 

Absence of toxin gene transfer from Clostridioides difficile strain 630∆erm to nontoxigenic C. difficile strain NTCD-M3r in filter mating experiments 

Dear Dr. Gerding:

I'm pleased to inform you that your manuscript has been deemed suitable for publication in PLOS ONE. Congratulations! Your manuscript is now with our production department. 

Kind regards, 

on behalf of

Dr. Ulrich Nübel 

Academic Editor

PLOS ONE